# Overview of the Cast Polyolefin Film Extrusion Technology for Multi-Layer Packaging Applications

**DOI:** 10.3390/ma16031071

**Published:** 2023-01-26

**Authors:** Damian Dziadowiec, Danuta Matykiewicz, Marek Szostak, Jacek Andrzejewski

**Affiliations:** 1Faculty of Mechanical Engineering, Poznan University of Technology, Piotrowo 3, 61-138 Poznan, Poland; 2Eurocast Sp. z o.o., Wejherowska 9, 84-220 Strzebielino, Poland

**Keywords:** multilayer, polypropylene, polyethylene, film, extrusion casting, packing

## Abstract

The review article presents the technology of producing polyolefin-based films by extrusion casting. Due to the wide use of this type of film as packaging for food and other goods, obtaining films with favorable properties is still a challenge for many groups of producers in the plastics market. The feedblock process and multimanifold process are the main methods of producing multi-layer film. In the case of food films, appropriate barrier properties are required, as well as durability and puncture resistance also at low temperatures. On the other hand, in order to properly pack and present products, an appropriate degree of transparency must be maintained. Therefore, processing aids such as anti-slip, anti-block and release agents are commonly used. Other popular modifiers, such as waxes, fatty acid amides and mineral fillers—silica, talc or calcium carbonate–and their use in film extrusion are discussed. The article also presents common production problems and their prevention.

## 1. Introduction

Polyolefins are a type of polymer or copolymers made from simple olefinic hydrocarbons and are one of the most numerous groups of polymers processed worldwide and in Europe in the packaging industry, including film production. Thermoplastics are mainly used, accounting for 84% of the plastics market [1]. In the case of multilayer packaging, the primary materials used are LDPE (low-density polyethylene), PP (polypropylene), HDPE (high-density polyethylene), and polyethylene terephthalate (PET) [2,3]. Materials made of foil must have different properties depending on the scope of their use. Specific requirements are set by the food and medical industries and are mainly related to low water vapor permeability, high flexibility and appropriate sealing parameters [4]. In addition, good mechanical and optical properties are essential for packaging food, textiles, fancy goods and vegetables or flowers [5,6]. The development of film materials with favorable mechanical and performance properties is still the subject of much research [7,8,9,10].

According to the report of Plastics Europe from 2021 packaging, including commercial and industrial packaging (40.5%) and the building/construction industry (20.4%), they are the largest end-use markets of plastics [11]. Additionally, the demand for polypropylene on the market is the highest and amounts to 19.7%, followed by polyethylene: PE-LD and PE_LLD 17.4%; PE-HD and PE-MD 12.9%; further polyvinyl chloride PVC 9.6% and polyethylene terephthalate PET 8.4% [11]. The main areas of application of polypropylene are food packaging, sweet and snack wrappers, hinged caps, microwave containers, pipes, automotive parts and banknotes.

Multilayer packages can be flexible or semi-rigid and contain polymer layers and inorganic layers [12]. Each layer has a specific function throughout the layout. The most commonly used food packaging materials are polyolefins, including polypropylene and polyethylene. Both polyethylene and polypropylene are characterized by adequate stability, flexibility, strength and chemical resistance. Furthermore, it should be emphasized that because of their easy processing, they can be successfully recycled. Moreover, due to its high thermal resistance, polypropylene can be successfully used for the production of dishes, bottles, canisters or laboratory equipment. Other advantages of the polypropylene group are low density, high melting temperature, chemical resistance and low manufacturing costs [13]. Polypropylene used in packaging usually acts as a moisture barrier due to its crystallinity and is utilized in the base or core layer, and is often modified to obtain the desired properties [6]. In addition, its high melting point makes the packaging resistant to higher temperatures. Oriented polypropylene (OPP) films are known for their inherent moisture barrier properties, while non-oriented polypropylene films only have a limited barrier function against water vapor and even less against oxygen or carbon dioxide [14]. For oriented films, these properties are significantly improved; however, they result in a higher production price, brittleness and strongly anisotropic mechanical properties and shrinkage. Therefore, non-oriented polypropylene films gradually gain importance over OPP due to lower manufacturing costs [15].

Polypropylene can be classified into three stereospecific configurations: isotactic, with methyl groups on one side of the polymer backbone; syndiotactic, where the methyl groups are switched on alternately on both sides; and atactic, characterized by an irregular arrangement of the methyl groups. Another classification concerns the proportion of monomers and specifies homopolymer (HPP) as containing only propylene monomer in a semi-crystalline solid form, random copolymer (RCP) containing ethylene (1–8%) as co-monomer in PP chains and impact copolymer (ICP) containing HPP and mixed phase of RCP [13]. Polypropylene is widely used in the production of various types of packaging and accounts for 16% of the entire plastics industry [16]. Homopolymer is the most commonly used type of PP in the production of plastic products. HPP contains both crystalline and non-crystalline regions, is rigid at room temperature, and has a high melting point but less transparency and reduced impact strength [13]. Due to the addition of ethylene, the RCP copolymer is characterized by favorable impact properties, lower melting point temperature and increased flexibility. Impact copolymers (ICP) are physical mixtures of HPP and RCP with a total ethylene content of about 6–15% by weight. Impact polymers can be used at low temperatures with higher impact resistance [13]. Various types of polypropylene are used in industry, depending on the properties the products made of it should have. The use of polypropylene for transparent products has recently been popular due to its low price and an acceptable degree of haze [17]. Low haze products are mainly made of RCP copolymer. Table 1 lists examples of the types of polypropylene used in foil extrusion and its properties. Due to the technological requirements, this polypropylene has an average melt flow index, which facilitates the film extrusion process.

For the production of packaging, polyethylene is used the most, followed by polypropylene [11]. There are three main types of polyethylene: low-density PE (LDPE) with a density 0.91–0.940 g/cm^3^, linear low-density PE (LLDPE) with a density 0.91–0.92 g/cm^3^, and high-density PE (HDPE) with density 0.941–0.967 g/cm^3^ [18,19]. Additionally, the following can be distinguished very low-density PE (VLDPE), ultra-low-density PE (ULDPE) and medium density PE (MDPE), and ultrahigh molecular weight PE (UHMWPE). Polyethylene is characterized by low strength, stiffness, and thermal stability [20]. HDPE is relatively stiff with low thermal stability and very low moisture absorption and can be applied in consumer bags or films. Moreover, it has excellent gas and water barrier properties as it has higher crystallinity than LDPE film. LDPE is semi-crystalline, translucent, with low density and hardness characteristics and relatively good insulation properties, and is used for the production of squeezable tubes and bottles, wrappers and bags and coating material for bottle cartons. LDPE flexible films are also applied for frozen foods, bakery products, fresh meat and poultry. In packaging production, polyethylene is mainly used for its good sealability and for producing a moisture barrier layer [21]. Above all, low-density polyethylene and linear low-density polyethylene (LLDPE) are applied as sealants, bonding layers, tie resins, adhesives or structural layers. Table 2 lists examples of the types of polyethylene used in foil extrusion and its properties.

The commonly used technologies in the production of films are casting technology and blow molding technology. Cast film extrusion allows for higher efficiency and greater uniformity in film thickness than in the blown film. Cast film extrusion takes place in the following steps: extruding the molten polymer through the slot die, then stretching it a certain distance through an air gap onto cooling rollers, and cooling and solidifying the molten film [22]. The chill-roll element is usually water-cooled or heated to a temperature range of 15 °C to 80 °C for quick cooling. Transferring the film to subsequent rolls ensures control of the sheet dimensions and its even winding [23]. Depending on the purpose of the extruded foil, various additional auxiliaries are used or combined with other materials [24]. Multilayer polymeric materials are formed to provide a gas barrier [25] or suitable mechanical [26] and optical properties [27] to the films [28].

Although the film extrusion process has been used on a large scale for many years, manufacturing multilayer products is still a challenge for a large group of plastic packaging manufacturers. Therefore, this article aims to characterize the processing methods and auxiliaries in the flat-film extrusion process. In particular, the review describes the structure of multilayer film and the function of individual layers in the package, the cast film extrusion technology by feedblock process and multimanifold process, technological problems in the manufacturing as well as additives and modifiers for polyolefin films.

## 2. Structure of Multilayer Packaging

Multilayer packaging can be divided into rigid and flexible as well as special-purpose and ordinary packaging [12]. Multilayer materials significantly extend the shelf life of fresh, packaged foods, mainly meat, fish, poultry and cheese [29]. According to Bauer et al., flexible multilayer materials account for as much as 10% of the weight of all packaging products [21]. The main functions of multilayer cast films are shown in Figure 1. Due to their properties, flexible films are used for packaging food and everyday products and for the protection of medical equipment.

The following polymers are mainly used for the production of multilayer films: high-density polyethylene (HDPE), low-density polyethylene (LDPE), linear low-density polyethylene (LLDPE), polypropylene (PP), polyamide (PA), polyethylene terephthalate, polystyrene (PS), ethylene vinyl alcohol (EVOH), polyvinylidene chloride (PVDC), ethylene vinyl acetate (EVA), polycarbonate (PC) and polyvinyl chloride (PVC).

This type of foil can contain from 2 to even 24 layers. Each layer fulfills a specific role in ensuring the desired properties of the product. In such a complex system, the following layers can be distinguished: a seal layer, a barrier layer, a tie layer, a structural layer, an outer layer and a coating [30]. Another division of the layers is the barrier, active and control layers. The characteristic arrangement of layers in multilayer packaging is presented in Figure 2.

The sealing layer is in direct contact with the packaged products; therefore, it must provide a barrier to the substance from the outside, e.g., moisture or air [31]. Consequently, it is most often made of polyolefins such as polyethylene or polypropylene. In order to protect the freshness of products and maintain all properties, special barrier layers are used. This layer acts as a barrier to external substances or provides good retention and protection for the active ingredients inside the packaging [32]. An oxygen barrier is used to inhibit the rancidity and the proliferation of aerobic microorganisms of the packaged goods [33]. Often, ethyl vinyl alcohol is used to improve the oxygen barrier in food or cosmetic packaging. Despite its high oxygen barrier, EVOH is a poor water vapor barrier. Therefore, multilayer packaging is usually used as an inner layer insulated from the outside with a protective layer, e.g., made of polyethylene [34,35]. To keep the food fresh, a light barrier layer such as an aluminum coating or a polymer filled with TiO_2_ is used [36]. In addition, many products use a migration barrier preventing plastic components from penetrating the food [37,38]. If the layers do not connect together sufficiently, tie layers can be used [39,40]. Various types of novel tie resins, graft copolymers, and maleated polymers are used as binders. In order to improve the stiffness and stability of the film, a structural layer consisting of a cheaper filler is used, which increases the thickness and puncture resistance of the package [2,41]. Different requirements for the processing of the film are achieved by the use of an appropriate external layer (most often PE or PET) susceptible to, e.g., printing. Out of the many types of flexible film packaging, only the special-purpose ones will require so many functional layers. In most cases, the films used to package the products may be multilayered, but for ease of production they are made of a single type of polymer, e.g., polypropylene or polyethylene. Multi-material multilayer plastic packaging poses a challenge to conventional recycling methods. It is assumed that the following will be developed in the coming years: highly efficient recycling of materials, recycling into hydrocarbons, simple extraction of raw materials or down cycling. Advanced, high-efficiency material recycling encounters systemic bottlenecks such as insufficient post-consumer waste sorting technology. In turn, chemical recycling requires significant investment in infrastructure and pre-segregation [42].

Due to its properties, polypropylene can be used as a moisture barrier and provides mechanical strength, and can also be covered with heat-sealable coatings. Therefore, films made of polypropylene are characterized by low water vapor permeability, high flexibility and very favorable seam parameters. Figure 3 shows an example of the use of polypropylene film for food packaging. Single-layer films are widely used for thermoforming in the food, pharmaceutical and technical industries.

Multilayer films are intended for packing products such as meat, cold cuts, cheese, dried fruit, delicatessen products, desserts and dairy products. Thanks to several layers of one type of polymer in packaging materials, it is possible to produce films with high durability and functionality. The multilayer polymer film often comprises at least one core layer sandwiched between two skin layers, as shown in Figure 4.

Barrier properties play a key role in the production and functionality of multilayer film packaging. Many reports analyze the dependence of the selection of individual layers on the film barrier [43,44]. As reported by Mueller et al., two effects are responsible for the permeation of gases and vapors through polymer films: gases and vapors pass through microscopic pores and the diffusion effect of the solution [45]. Gases and vapors dissolved on the surface of the polymer diffuse through the polymer in a concentration gradient and then evaporate from the other side of the polymer. The two primary techniques for assessing film barrier properties are Oxygen Transmission Rate (OTR) and Water Vapor Transmission Rate (WVTR). The main factors influencing oxygen permeability are polymer characteristics such as chain structure, degree of crystallinity, formulation, processing properties, and physical interaction between penetrant and barrier material [46]. In order to improve the barrier properties of PP and PE films, they are combined in the co-extrusion technology with other polymers. Common barrier materials are ethyl vinyl alcohol (EVOH), Polyvinylidene dichloride (PVCD), and polyamide (PA), which can be incorporated into layered systems to improve their properties and functionality. Figure 5 shows a five-layer polyethylene film with a barrier layer made of ethyl vinyl alcohol (EVOH). The two thickest outer layers are polyethylene, then two layers of tie based on LLDPE modified with maleic anhydride and EVOH as the core. The permeability properties of EVOH copolymers result from the ratio of ethylene and vinyl alcohol in the copolymer. Furthermore, barrier properties can be provided by the use of an aluminum foil or a metalized aluminum layer on a PP or PE polymer film [47]. Mostly laminates based on Al metalized biaxially oriented polyethylene terephthalate (BOPET) and biaxially oriented polypropylene (BOPP) are used as barrier layers in food packaging materials [48].

Mechanical properties, and adequate strength, which ensures resistance to punctures and splitting, are provided by threads made of PE or PP with a thickness of up to 70% of the total thickness of the film. Co-extrusion in such a structure requires a lot of adhesives for interconnecting layers. Table 3 summarizes an example of a layer arrangement in a packaging film. Fresh produce packaging film can consist of four to seven layers. The lower the concentration of ethylene, the lower the oxygen permeability [49]. Other methods of improving the barrier properties of films include the use of additional layers of proteins [50,51], copolymers [52,53] or nanoparticles [54,55].

Dorey et al. proved that the gamma irradiation process of a PE/EVOH/PE multilayer film significantly improves the oxygen barrier properties of the film [56]. Multilayer film contains two layers of LLDPE as an external and contact layer and one internal layer of EVOH. Zhang et al. investigated the confined crystallization effect of high-density polyethylene (HDPE) in multilayer films [57]. He used a new type of cyclic olefin copolymer (HP030), producing multilayer films of HDPE/HP030 with 33, 65, 129 and 257 using a forced-assembly, layer multiplication co-extrusion process. Furthermore, multilayer films of PP/PC with 33, 65, 129 and 257 alternating layers were also fabricated by the same method. The authors proved that both HDPE/HP030 and PP/PC films indicated enhanced for oxygen and WVTR barrier properties as a result of the process of confined spherulite morphology, which increases the tortuosity for gas diffusion. Decker et al. described the process of layer multiplying co-extrusion as a method of producing gas barrier films consisting of alternating layers of low-density polyethylene and maleic anhydride grafted linear low-density polyethylene (LLDPE-g-MA)/organoclay nanocomposite [58]. The multilayer nanocomposite system reduced the oxygen permeability by 63% compared to the unfilled LDPE/LDPE-g-MA system. Sanchez-Valdes et al. investigated a layer of polyethylene–silver nanoparticles in a five-layer barrier film. The five-layer film produced in the co-extrusion technology consisted of the following layers: PE as an external layer, PA6 as the central layer and PE-g-MA as tie layers (PE/tie/PA6/tie/PE) [59]. A thin layer of silver-PE nanocomposite was applied to the 5-layer film using lamination, casting and spraying. The spray method showed the most favorable biocidal properties. In addition, antimicrobial agents can be used to improve the bacterial resistance of the package. Ha et al. applied grapefruit seed extract on the surface of a multilayer polyethylene film in contact with food by co-extrusion or a solution coating process [60], whereas Marcus et al. incorporated TiO_2_, and ZnO nanoparticles into low-density polyethylene films to improve the anti-bacterial properties of the package [61]. In addition, when designing a barrier package, the actual durability of the food in contact with the functional barrier and the minimum thickness of the functional material should be taken into account to ensure adequate barrier properties [62].

## 3. Methods for Producing a Multilayer Film

According to Gholami and others, the global market for multilayer polymer films (MLP) in 2017 was $5.50 billion and is expected to reach $9.58 billion by 2026 [63]. The ease of packing many products in foil to protect them generates such high production of these materials, and thus the number of methods and additives used in technological processes is constantly increasing. The properties of multilayer films depend largely on the production method, which may include the following: methods based on polymer solutions and methods based on polymer melts. During the production of foil from a polymer melt, rheological factors play a key role [64], such as the following: slip in shear fields [65], hardening in extensional fields [66], viscoelastic instabilities [67], layer breakup [68] and variety of relaxation times in polymers [69]. The selection of the appropriate production technology, polymer material and auxiliaries allows for producing high-performance multilayer films with specific properties [70]. Extrusion film casting or blown film extrusion can be used to produce thin thermoplastic polymer film. Both processes are very efficient and enable the production of high-quality films with a specific thickness and transparency.

An important issue is to design the production system to obtain a film of uniform width and thickness. Therefore, the degree of stretching of the film in the air gap is quantified as the draw ratio (DR), which is defined as the tangential velocity of the foil on the chill roll divided by the linear velocity of the stop exiting the die [71]. The difficulties that appear during the production of foil are mainly two phenomena: the occurrence of necking, i.e., inhomogeneous reduction in the width of the film, and edge-beading, i.e., an increase in the thickness of the film at its edges. In industrial extrusion, the edges of the film are often trimmed to remove edge beads. In turn, a co-extrusion process is used to produce the multilayer film.

Moreover, co-extrusion reduces the costs associated with multistage lamination and coating processes, where specific layers have to be produced separately and then primed, coated and bonded [72]. In the process of co-extrusion of the cast film, a matrix with a flat geometry allows the material to be formed through a narrow slit. Two types of die geometry can be distinguished: multi-manifold dies and single-manifold die with a feedblock [73]. The co-extrusion system is made of parts that take a polymer melt of two or more extruder, mold the melt and then deliver it to the point of connection where they are melted, stacked and formed into a finished product of a specific width and thickness [74]. The system consists of the following components: an adapter, a feedblock and a single- or multi-cavity die. The adapter collects and traces different melt streams to the feed block during multilayer material production. Adapters connect the extrusion system and the co-extrusion system and can be standalone or integrated with feedblocks or dies. A variety of adapters are available with fixed flow channels or allowing multiple flow paths where the polymer melts between the adapter inlet and the adapter output to the feed block and/or die. The feedblock shapes and adheres the multiple layers of polymers exiting the adapter into the polymer stack for delivery to a flat die. The geometries and feedblock combinations can be distinguished as follows: fixed geometry feedblocks which include the segmented flow of the Dow feedblocks and the stepwise addition of the Welex modular designs with interchangeable cassette inserts and the variable geometry feedblock such as the Cloeren style feedblock with adjustable vane designs [74]. Figure 6 shows examples of feedblocks geometries. Additionally, the dies have been modified with simple forms to the modern co-extrusion dies.

The following dies can be distinguished: the single-manifold die, combined with a feedblock, and the multimanifold co-extrusion die. The feedblock method is cheaper to the application than the multimanifold for the reason that melt streams travel some distance before reaching the die exit, and irregular flow patterns can develop at the interface of the different melt streams [75]. This is particularly important when attempting to coextrude melts of significantly differing viscosities. Lower-viscosity materials may encapsulate the more viscous materials. The optional solution is to keep the melt streams separated until just before the die exit for the multimanifold construction. Whereas multi-manifold gives an opportunity for the coextrusion of plastic materials with significantly differing viscosities.

### 3.1. Feedblock Process

Cast film extrusion of a multilayer film usually includes the following stages: dosing the material to the plasticizing system of individual extruders, plasticizing at a temperature adequate to the polymer, passing the material through a filter and pump, then entering the feedblock where the layers are joined and going to the pouring nozzle, and then to the cooling rotating roller. Different types of polymer materials can be combined using feedblock technology with a multilayer die and extruded onto a chill roll. Moreover, the corona treatment station is installed in the extruder system to increase wettability and adhesion film. Additional equipment for the line can also be a system for measuring the thickness, checking the gauge profile, cutting section for removing the edges and a winder. The plastic scrap from cutting rolls that is produced during extrusion is usually recycled by grinding and recirculation into the process [76]. The system for the industrial extrusion of a three-layer film is shown in Figure 7.

In the case of the production of polypropylene and polyethylene film, the thickness of the individual layers should be carefully selected. The outer layers produced in extruders 1 and 3 are usually 15% of the thickness in relation to the entire film and consist of a copolymer and/or homopolymer with anti-blocking additives, ensuring adequate surface tension and sealing properties. The core layer may be made of copolymers and/or homopolymers, ensuring adequate strength, and makes up about 70% of the film thickness. A number of layers, melt ratio polymers and the final film thickness can be independently controlled to produce a product with intended properties and layer thicknesses in the micro-scale [73]. Another important factor is choosing the right temperature for each chill roll. Chill roll 1 and chill roll 2 are cooled while the successive rolls are at ambient temperature. After the first two rolls have passed, the thickness of the foil is measured, and then the foil is wound onto a metal roller. Creating a line for the continuous extrusion of film requires a large warehouse space for raw materials and products and a working area for the operation of interconnected machine components. The industrial cast three-layer film extrusion line system with automatic raw material feeders are presented in Appendix A. The extruders are located on the upper level, and the chill-roll system below ensures adequate heating or cooling and transport of the finished product. The device responsible for the preparation of the roll is a winder which collects the cooled material from chill rolls (Appendix A).

Difficulties in using polypropylene and polyethylene to make foil are its susceptibility to oxidation, thermo-oxidation, and ultraviolet degradation [77,78]. To eliminate this problem, plastic producers prepare mixtures with appropriate stabilizers. The most common degradation phenomenon during extrusion is the thermo-oxidation process. Under the influence of high temperatures, oxygen is involved in the autocatalytic action-reaction of shortening polymer chains to obtain volatile products of molecular weight. When designing an extrusion line, it is necessary to take into account, first of all, the types of extruded material, the maximum width of the filament roll, the number of layers and the assumed production quantity [78]. In summary, the main cast film line components are gravimetric feeding systems, extruder, filtration systems, flat die system, cooling unit, automatic gauge control systems, corona treatment, winder and control system.

### 3.2. Multimanifold Process

Additionally, films with thinner layers may be fabricated by nanolayer coextrusion with layer multiplying elements. The innovation of this co-extrusion process is a combination of the conventional co-extrusion of two or more polymers into a layered feed block with an additional layer multiplication by a series of multiplier matrices [79,80]. This technology enables the production of films containing tens of up to thousands of layers with individual layer thicknesses from micro to the nanoscale. Multilayer co-extrusion relies on the viscoelastic properties of shear-melting polymers to produce a layered structure. Figure 8 shows a two-component multilayer co-extrusion system. A and B extruders extrude two distinct types of polymers. A melt pump is installed on each extruder to control the relative volume composition of polymer A and B which is proportional to the ratio between the layer thickness of the two polymers. From the feedblock, the two materials layers fill in a series of layer multiplication dies, each of which doubles the number of layers through a process of cutting, spreading and stacking the viscoelastic melt [79]. At each multiplier, the layered molten polymer is divided vertically into two parts, then one part flows and spreads in the upper channel, and the other part flows and spreads in the lower channel to stack (Figure 8). The layer number is defined by the number of the multipliers added and calculated as 2^(n+1)^ for the A/B structure and 2^(n+1+1)^ for the A/B/A structure.

## 4. Technological Problems in the Production of Polyolefin-Based Multilayer Films

Another important issue during large-scale industrial film production is the quick elimination of errors and defects during continuous production. For this purpose, additional equipment is used, such as foil thickness gauges or electrostatics pins, to ensure proper adhesion of the foil to the chill roll (Figure 9a,b). The quick response of the operator and the introduction of corrective actions allow for avoiding material losses and reducing additional production costs. Table 4 lists the most common technical problems and the method of solving them.

These problems can be solved immediately in the production line, mainly by selecting the appropriate extrusion temperature or chill rolls. Common problems in the process are an inadequate design of peripheral devices, material defects, a problem with plasticization of a large batch of material, poor mixing, moisture release, local overheating of the material or contamination [81]. Figure 10 shows typical mechanical defects occurring during extrusion, such as creases and local folds on the edges of the roll. They are the result of an inappropriate selection of chill-roll operating parameters, such as temperature and winding speed, whereas all sharp elements on the film die, rollers or winders will cause a film defect such as scratches or discontinuities of the film (Figure 11a). On the other hand, instabilities in the extrusion flow rate and the speed of pulling the roller or winder will change the thickness and width of the film. The defects resulting from flow instability are shown in Figure 11b. The molten polymer is an incompressible fluid, the film web is stretched in the machine direction, and its thickness and width are reduced. Additionally, there are reductions in thickness, and width is function of the stretch factor and the length of the drawing zone [82]. Reducing the width of the film is defined as the “neck-in” phenomenon, whereas edge stress effects arise due to differences between stress and strain conditions at the center and at the edge of the film [83].

According to the data [84], more than 100 million tons of multilayer thermoplastics are manufactured each year globally. Unfortunately, as much as 40% of multilayer films are not used for packaging due to ineffective technological processes used in their production [85]. Therefore, solving problems that arise during a continuous production process is crucial to improve efficiency and to reduce material waste. In the case of film production from one type of polymer, i.e., polypropylene, primary recycling is carried out on the production line, including recycling the processed raw material.

Dies and feedblocks are a significant aspect of the cast film extrusion process due to their layering and forming function, which determine the manufacturing efficiency and properties of the final product. A feedblock, connected multiple polymer streams into a narrow multi-layer that is distributed to full width in a single-manifold die. Therefore, constructors design new solutions, such as precision feedblocks that can be connected directly to the extruder and die without additional parts, thus saving costs. Additionally, the percentage of each layer can be adjusted on line in a larger range. For example, in vane-type feedblock, the extrusion of different layers occurs, only by easily changing the split-flow bars, without dismantling the whole set of feed block. In insert-type feedblock the flow channel is constructed from a pack of inserts to achieve the best uniformity of coextrusion. Moreover, during changing the layer composition, the installing inserts with flow channel or normal block is only required. In turn, the multimanifold method ensures higher layer uniformity and thickness accuracy in that it eliminates much of the layer-interface deformation that occurs when multiple layers pre-assembled in a feedblock and are then spread simultaneously through the die.

## 5. Additives and Modifiers of Casting Film

Many functional additives or processing aids are introduced into thermoplastics, including catalysts, antioxidants, thermal stabilizers, plasticizers and dyes which help to extend the material’s useful life, most often in small amounts not exceeding 2% [86]. The most commonly used antioxidants are sterically hindered phenol (e.g., butylate hydroxytoluene (BHT) or 2,6-Di-tert-butyl-4-[[(octadecyloxy)carbonyl]ethyl]phenol) or organic phosphite (e.g., trisnonylphenyl-phosphite (TNPP) [87]. Additionally, systems based on hindered phenols with other additives provide long-term thermal stabilization of polypropylene [88]. Calcium stearate, magnesium stearate, cobalt stearate, and zinc stearate have to stabilize and process aid effects [89,90,91]. Furthermore, Bensaad et al. proved that calcium stearate might act as a pro-oxidant agent on the degradation of polypropylene under natural exposure. According to Zatloukal et al., polymer processing additives (PPAs) are a group of additives used in single-layer extrusion to reduce the appearance of defects [92]. In turn, G. Wypych divides the polymer additives into anti-blocking, release, and slip. Release agents are most often used in the molding industry as well as in calendering and extrusion, where separating the product from the mold requires the use of special substances. The introduction of an anti-blocking compound aims to reduce forces of adhesion between the materials in contact. The slip agent must compensate for the material shortage due to too much friction between the two contact surfaces [93]. The slip additives are popularly applied to control the friction between film surfaces, which is crucial in controlling packaging production speed [94]. These modifiers are available in premixes in a concentration of 5–10% in pellets and then introduced during film production at a lower concentration. Initially, they are found in the mass of the polymer and then migrate to the surface of the film.

During the production of films from polymers such as polypropylene or polyethylene, static interactions occur, which require the use of external anti-blocking agents or in the form of a masterbatch [95]. Anti-blocking is the term used to define measures that prevent foil sheets from sticking together. Polyolefin films tend to stick together due to strong van der Waals interaction or electrostatic charges during close contact. With increasing temperature, pressure and processing time, the tendency to stick increases [96]. In turn, the task of the slip agent is to supplement the shortage of material associated with too much friction between two surfaces of contacting material [93].

Anti-block modifiers include waxes, fatty acid amides, silicones and silica and silicates. Additionally, such mineral fillers as calcium carbonate, kaolin, talc and zeolites are commonly used anti-block agents. Popular slip agents include waxes, silicones, secondary amides, fluorocompounds, fatty acid amides, fatty acid esters, fatty acid salts and fatty acids. It can be seen that in some areas the same substances can be used as both anti-blocking and slip agents. Waxes are most often petroleum-derived consisting of complex mixtures of paraffinic, isoparaffinic and naphthenic hydrocarbons. They are characterized by a microcrystalline structure, which makes them quite flexible and thin and can act as an anti-block agent or improves film clarity. Non-polar waxes do not mix with most polymers, so they quickly migrate and remain on the surface of the material. Fatty acid amides can be referred to as slip agent or anti-blocking agents that reduce friction and adhesion forces, form a thin layer on the surface of the finish products for an excellent glide effect, improve filler dispersion, high film clarity or color stability. One of the popular fatty acid amides is Erucamid, which reduces blocking, or the tendency for the layers to stick together and to facilitate separation [97]. Wakabayashi et al. described an application of slip agents such as behenic acid (docosanoic acid) in a polypropylene film, erucamaide (13-cis-docosenamide) in ethylene copolymerized polypropylene film and the use of molecular dynamics to predict the self-association of slip agents in the iPP film [98]. The use of fatty acid amides is preferred over the use of inorganic materials as blocking agents. The difficulties associated with introducing fine particles ~10 µm solids into conventional dosing equipment, such as dusting or agglomeration, are reduced for the most part. Moreover, in many cases, the polymer film becomes somewhat hazy and has lower transparency and lower gloss when using inorganic powders as an anti-blocking additive [99]. Unfortunately, organic compounds such as erucamide and oleamide provide good slip performance but may migrate from the film surface [100].

Silicones or polysiloxane are most often built of an inorganic silicon–oxygen backbone chain with two organic groups attached to each silicon center [101]. They are mainly used as modifiers or lubricants. Therefore, Yilgor et al. used silicone copolymers such as a poly-dimethylsiloxane-b-polycaprolactone triblock copolymer (PCL-PDMS-PCL) and a polydimethyl-siloxane-urea (PSU) segmented copolymer to modify the polypropylene surface [102]. The silicone copolymers used successfully improved the extrusion efficiency. Importantly, silicone-based concentrates dedicated to biaxially oriented BOPP polypropylene are already available on the market, allowing lower coefficient of friction (COF) and optimizing packaging production efficiency by enabling excellent printing and metallization [36].

Mineral fillers such as talc or silica can be successfully used as anti-blocking additives. Mineral materials have hydroxyl groups on the surface which can react with many polymers. It can be assumed that the low energy of hydrogen bond formation allows for their easy breaking and reforming, which slows down the migration of substances that can bind hydrogen [93]. Essche et al. described the use of silica with a higher pore volume to provide increased anti-blocking efficiency for films of low-density polyethylene and polypropylene 35 µm thick [103]. Studies proved that this type of silica reduces blocking strength, especially at lower levels of an anti-blocking agent of 1000–1500 ppm. The following forms of synthetic silica, such as micronized gel, fume, and precipitated silicas, are used as antiblocking agents. Various types of precipitated silicas are commercially available, and introducing a small amount of this material into the film <0.5% is sufficient to significantly reduce COF and obtain an anti-blocking effect [104]. Unfortunately, the inclusion of precipitated silica in the polymer affects the haze and reduces the transparency of the film. However, Evonik company offers a new type of silica dedicated to the production of foil with minimal influence on its haze. In turn, Espinosa et al. described the use of talc in a concentration of up to 5% *w*/*w* to produce polypropylene blown film [105,106] for use in food packaging. Low price, low moisture absorption and limited effect on haze and transparency make it widely used as a modifier and anti-blocking agent [107]. On the other hand, calcium carbonate is a mineral with hydrophobic properties and a well-controlled particle size. Its addition to the material eliminates the need to introduce anti-blocking additives. Zatloukal et al. characterize a group of materials based on fluoropolymer materials, most often incompatible with other polymers [92]. During the flow during extrusion, particles migrate to the die wall due to phase separation forming a thin layer of fluoropolymer. Therefore, slip is induced at the fluoropolymer/polymer interface, which reduces stress on the die wall and increases melting speed in this region. The use of fluoropolymers makes it possible to reduce the occurrence of such disadvantages as shark skin, die drool phenomenon, and degradation occurring during film production. In addition, phenolic-based antioxidants most often protect the finished product during use. On the other hand, antioxidants based on phosphites or thioesters protect the polymer during processing [108]. In turn, UV stabilizers which protect from UV radiation can be divided into UV absorbers which absorbing UV radiation and dissipating it and UV inhibitors which inhibit degradation of the polymer and are the most effective for polyolefins. For aesthetic reasons and to limit food spoilage, antifog masterbatches are used in foil packaging. Additionally, tie-layer resins based on anhydride-modified polyolefins are used to bond different polymers together in multilayer structures. Table 5 shows examples of commercial modifiers used in processing polyolefin-based films.

## 6. Conclusions

The use of extrusion casting technology to produce multilayer films is efficient and allows to obtain of a product with specific functionality and quality. Moreover, the introduction of anti-blocking, slip and release agents into the polymer matrix significantly facilitates processing, which allows for avoiding the production of defective films and reduces material losses during large-scale production. Due to the low cost, chemical inertness, good processability, nontoxicity, flexibility and recyclability of polyolefins are still widely used in the preparation of thin mono or multilayer films. The prepackaged food industry spans the world. To meet supplier expectations and keep food fresh for as long as possible, a variety of plastics are used to make disposable or reusable packaging. Due to its affordable price and processing that does not require complicated equipment, polypropylene is widely used to produce films on world markets. The greatest use of polypropylene and polyethylene is observed in injection molding technology, followed by extrusion and thermoforming. The film’s required barrier properties are obtained using additional coatings or multilayer structures.

## Figures and Tables

**Figure 1 materials-16-01071-f001:**
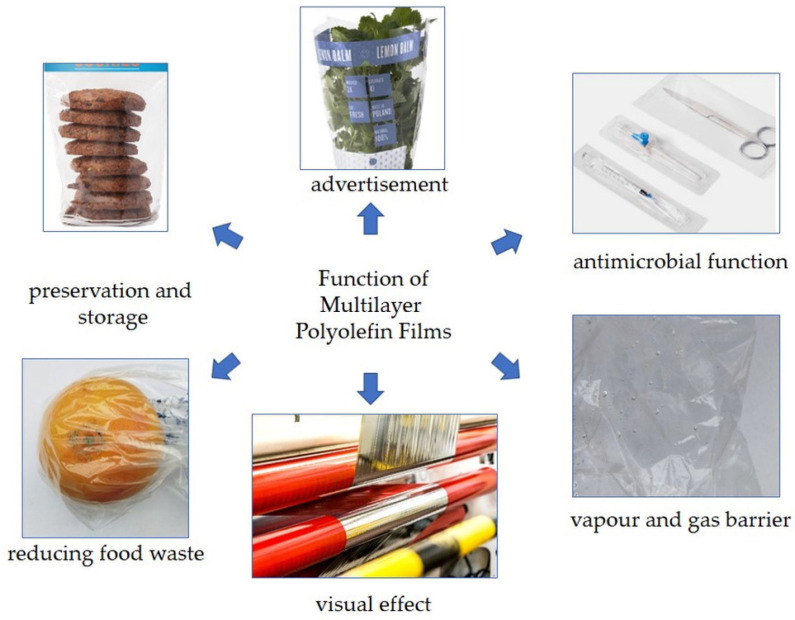
Features and applications of multilayer cast films.

**Figure 2 materials-16-01071-f002:**
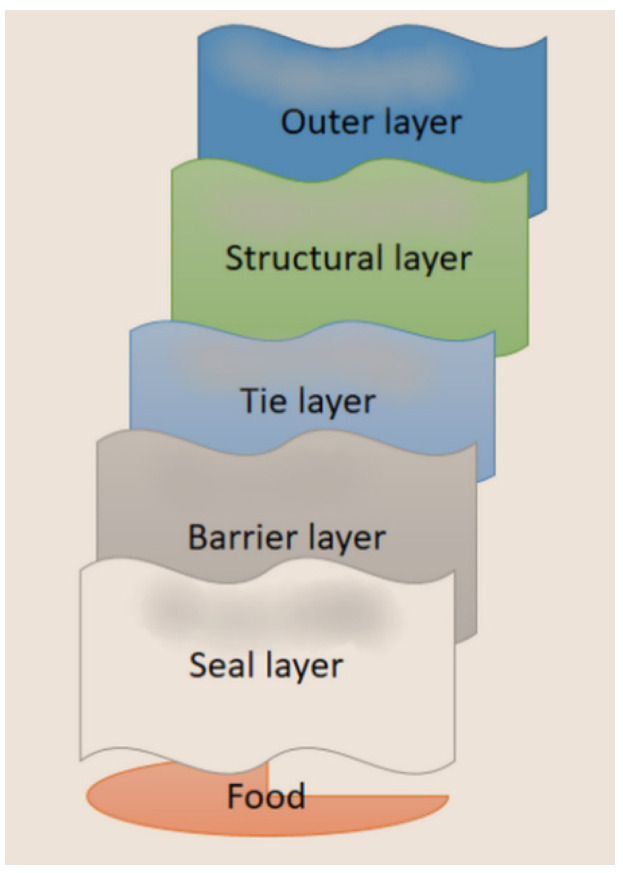
An example of the arrangement of layers in a multilayer film.

**Figure 3 materials-16-01071-f003:**
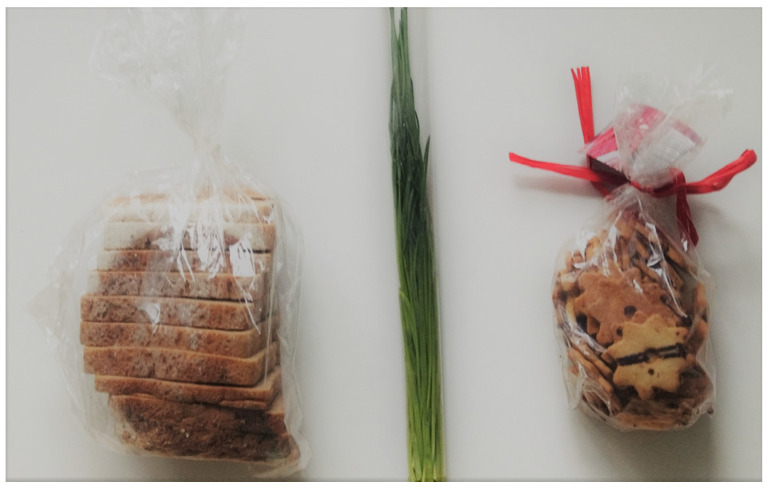
Examples of the use of multilayer film for food packaging.

**Figure 4 materials-16-01071-f004:**
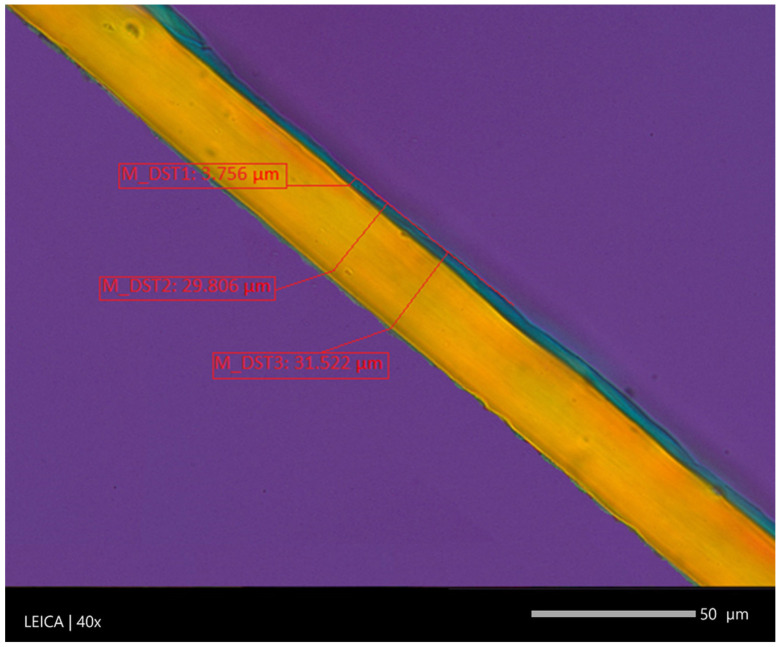
The characteristic structure of the three-layer cast polypropylene film, M_DST1, M_DST2, M_DST3 average layer thickness.

**Figure 5 materials-16-01071-f005:**
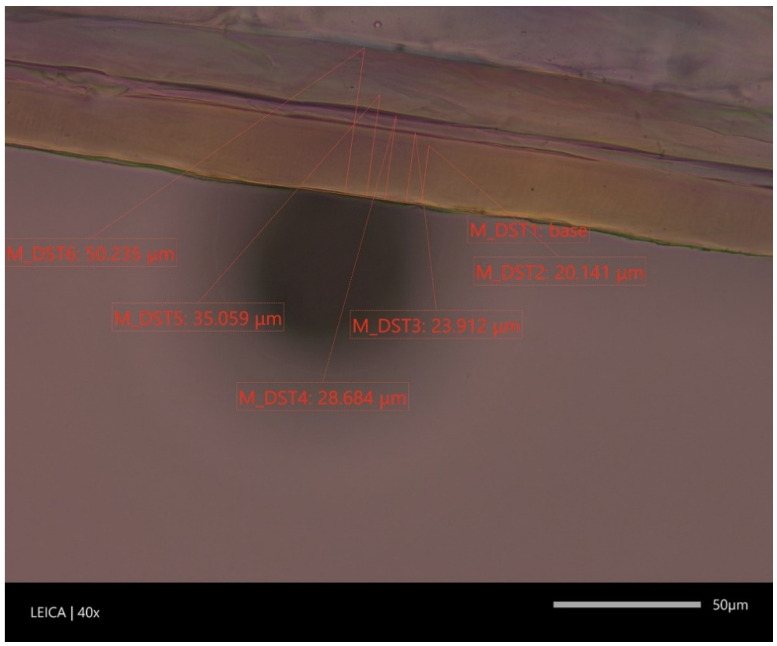
The characteristic structure of the five-layer cast polyethylene film with an internal ethyl vinyl alcohol barrier layer, M_DST1, M_DST2, M_DST3, M_DST4, M_DST5, MDST_6- average layer thickness.

**Figure 6 materials-16-01071-f006:**
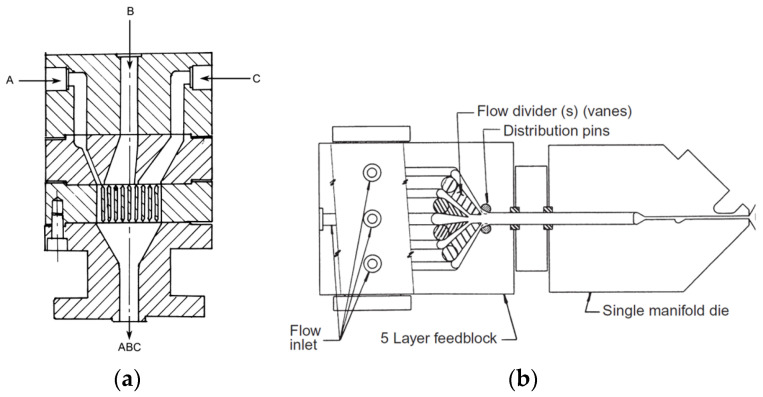
Feedblock with various geometries: (**a**) Three-layer Dow feedblock showing segmented flow plates A,B,C-material inlet; (**b**) Cloeren five-layer adjustable vane feedblock. (Reprinted with permission, Copyright 2016 Elsevier) [74].

**Figure 7 materials-16-01071-f007:**
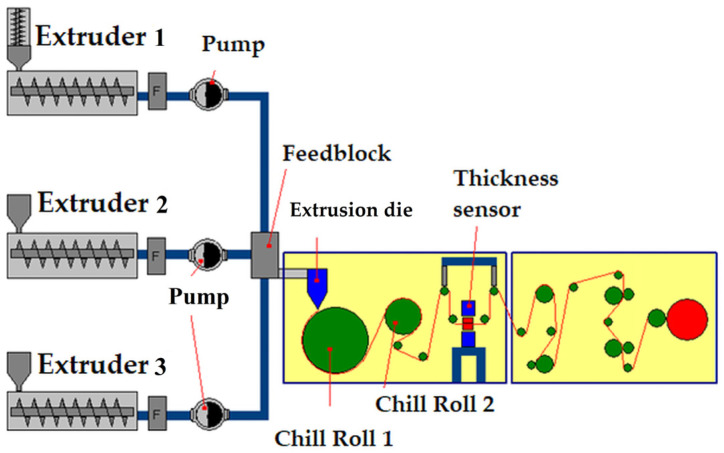
Scheme of an industrial line for extrusion of a three-layer polypropylene film, F-filter.

**Figure 8 materials-16-01071-f008:**
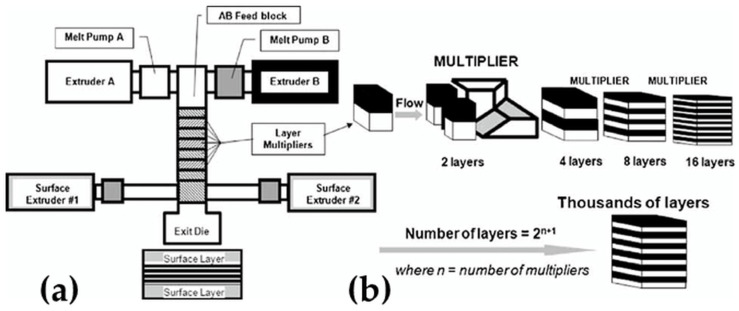
(**a**) Two component multilayer system comprised of extruders, pumps, feedblock, multiplying dies, surface layer extruders and exit die. (**b**) Schematic illustration of layer multiplication by cutting, spreading and recombining the melt stream. (Reprinted with permission, Copyright 2010 John Wiley and Sons) [79].

**Figure 9 materials-16-01071-f009:**
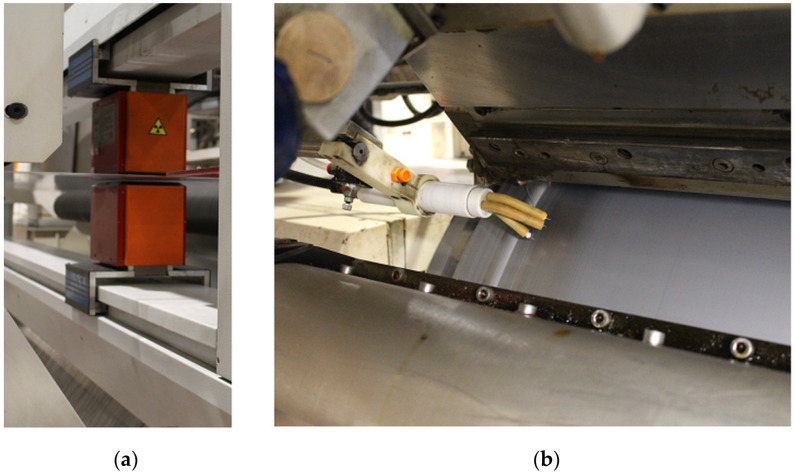
Extrusion line equipment: (**a**) thickness gauge and (**b**) extrusion die pouring plastic on the chill roll and electrostatic pins.

**Figure 10 materials-16-01071-f010:**
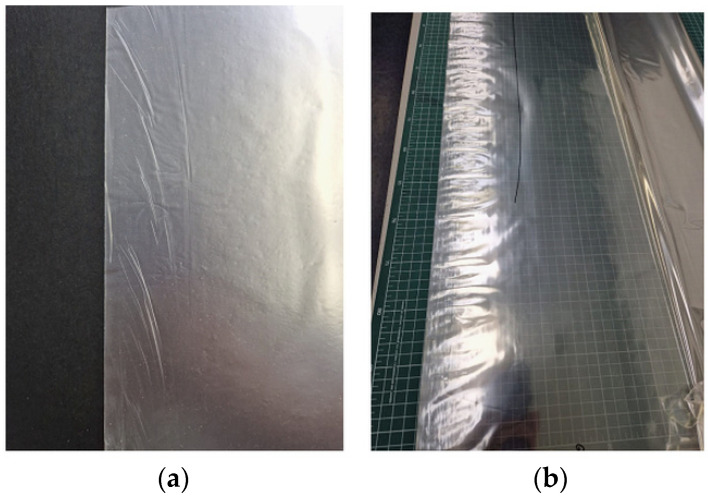
Typical mechanical defects occurring during extrusion, such as the following: (**a**) creases, (**b**) local folds on the edges of the roll.

**Figure 11 materials-16-01071-f011:**
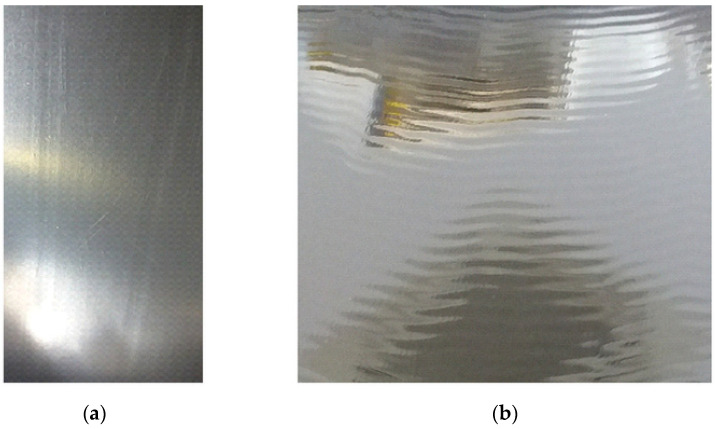
(**a**) Scratches on the film, (**b**) instabilities in the extrusion flow.

**Table 1 materials-16-01071-t001:** Characteristics of commercial polypropylene used in the production of the film.

Property	Moplen EP310D	PPH 6080	Borealis RD364CF	Braskem RP 225M
Melt flow index (230 °C/2, 16 kg), [g/10 min]	0.95	9.0	8.0	8.0
Vicat softening point (VST/B, 50N), [°C]	150	152	138	134
Film Tensile Strength at Yield, ISO 527-3, D 882, [MPa]				
-MD	16.0	23.0	40.0	20.0
-TD	16.0	23.0	30.0	19.0
Type	impact copolymer	homopolymer	random copolymer	random copolymer
Application	cast film extrusion	blown film	cast film extrusion	blown, cast film extrusion

**Table 2 materials-16-01071-t002:** Characteristics of commercial polyethylene used in the production of the film.

Property	Malen E FGNX 23-D022	Sabic 6318B LLDPE	Borealis FT7230	Braskem EB 853 LDPE
Melt flow index (190 °C/2, 16 kg), [g/10 min]	1.95	2.8	4.0	2.7
Vicat softening point (VST/B, 50N), [°C]	90	99	91	89
Film Tensile Strength at Yield ISO 527-1, 3, [MPa]				
-MD	20.0	13.0	20.0	27.0
-TD	19.0	10.0	18.0	20.0
Type	low density polyethylene	hexene copolymer linear low density polyethylen	low density polyethylene	low density polyethylene
Application	cast film extrusion	cast film extrusion	blown film extrusion	blown film extrusion

**Table 3 materials-16-01071-t003:** Common layer structure in multilayer film. Based on [49].

Structure	Composition [%]	Number of Layer	Function	Application
PP/PP/PP	15/70/15	3	non-barrier	Bags
LDPE/LDPE/LDPE/ LDPE/LDPE	26/15/16/15/28	5	non-barrier	Bags
LDPE/Tie/EVOH/Tie/LDPE	30/12/10/8/40	5	barrier	Meat Bags
LLDPE/HDPE/Tie/EVOH/Tie/HDPE/LLDPE	20/20/5/10/5/20/20	7	barrier	Meat Bags

**Table 4 materials-16-01071-t004:** Examples of the main technological problems in the production of polyolefin-based multilayer films.

Technological Problem	Solution Method
Scratches	Rolls not turning: Check the roll speed and adjust accordingly Check the balance
Low MD Elongation	Extrusion temperature: Increase extrusion temperature
Layers delamination	Use the resin with a similar MFR
Low MD Tear Strength	Use a resin with a lower density Increase the chill roll temperature Decrease the extrusion temperature
Low Stiffness	Use a resin with a higher density Increase the chill roll temperature

**Table 5 materials-16-01071-t005:** Additives and modifiers for polyolefin-based films.

Name/Producer	Type	Function
Tospearl™ 145FL (Momentive, United States)	4.5-micron silicone resin	anti-blocking agent
SPHERILEX 30 AB (Evonik, Germany)	precipitated amorphous silica	anti-blocking agent improved moisture vapor barriers
Constab AB 06001 PP (Kafrit Group, Germany)	10% synthetic silica (5 µm) in homopolymer	anti-blocking masterbatches
CALCIPORE^®^ 80T AL. (Reverte, Spain)	ultramicronized, treated calcium carbonate	anti-blocking agent, production of breathable film
Crodamide™ ER (Croda, United Kingdom)	erucamide	slip and anti-blocking agent
Chemstat^®^ HTSA #18–20M (PCC, Germany)	oleyl palmitamide	slip and anti-block, antistatic, mold release
Dynamar™ FX 5911X (3M, United States)	fluoropolymer	reduced die lip build-up, better gauge control
Ionphase™ fSTAT (Croda, United Kingdom)	proprietary	anti-static additive
Linanox 1010 (Linchemical, China)	phenolic antioxidant	antioxidant
ThermProtect 1001265-N (Ampacet, United States)	phenolic and phosphite antioxidents	antioxidant
Linsorb 236 (Linchemical, China)	benzotriazole	UV absorber
SABOSTAB^®^ UV 81 (SABO S.p.A., Italy)	benzophenones	UV absorber
SABOSTAB^®^ UV 94 (SABO S.p.A., Italy)	hindered amine light stabilizer	UV light stabilizers
LOWILITE™ GR 6294 (SI Group, United States)	hindered amine light stabilizer	UV light stabilizers
Ampacet Antifog (Ampacet, United States)	masterbatches	anti-fog additives
Atmer™ 110 (Croda, United Kingdom)	ethoxylated sorbitan ester	anti-fog additives
Plexar PX2600 (LyondellBasell, Netherlands)	chemically modified resins	tie-layer adhesive
AFFINITY^TM^ KC8852G (DOW, United States)	ethylene-octene copolymer	polyolefin plastomer
ZELAS™ R-TPO (Mitsubishi Chemical Corporation, Japan)	olefin-based thermoplastic elastomer	polyolefin elastomer

## Data Availability

Not applicable.

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
