# Peer review of "Overview of the Cast Polyolefin Film Extrusion Technology for Multi-Layer Packaging Applications"

_materials, 2023, doi:10.3390/ma16031071_

Round 1
Reviewer 1 Report
In this review article the structure, manufacturing process, problems and additive of multilayer packaging materials based on polyolefins are discussed. In general, this is a fairly well written review. The following minor issues should be addressed.
1. Title should be revised in order to be more representative, e.g. the term “multilayer packaging ” should be included.
2. Rewrite abstract. At some points it is written as regular and not review article e.g. “the process was fully characterized” or “the following modifiers…..were charactrized”.
3. References should be included in Table 3 in page 8 (there is a second table 3 in page 13).
4. Figures 8, 9 and 10 are not really informative. I suggest to present them as supporting information.
5. Please enrich Table 4 in page 17 with more additives e.g. antioxidants, compatibilizers etc.
Author Response
Dear Reviewer 1,
Thank you for your review, which will improve the quality of our article. The manuscript has been revised as recommended and all changes to the text are marked in red. Moreover, the English language was checked by a native speaker.
In this review article the structure, manufacturing process, problems and additive of multilayer packaging materials based on polyolefins are discussed. In general, this is a fairly well written review. The following minor issues should be addressed.
- Title should be revised in order to be more representative, e.g. the term “multilayer packaging ” should be included.
Answer: The title has been corrected as recommended
- Rewrite abstract. At some points it is written as regular and not review article e.g. “the process was fully characterized” or “the following modifiers…..were charactrized”.
Answer: The abstract has been revised as recommended
- References should be included in Table 3 in page 8 (there is a second table 3 in page 13).
Answer: References have been added, the table has been prepared partly from the source, partly from the authors' own data.
- Figures 8, 9 and 10 are not really informative. I suggest to present them as supporting information.
Answer: The photos have been removed and moved to supplementary files
- Please enrich Table 4 in page 17 with more additives e.g. antioxidants, compatibilizers etc.
Answer: Table 4 has been supplemented with additional modifiers as recommended
Reviewer 2 Report
This review paper describes the overview of polyolefin film extrusion technology. This paper contains more textbook content than an academic paper. However, this paper is thought to be helpful to researchers in the field of polymers because it conveys industrially meaningful content. Therefore, I agree with the publication of this paper. However, a prior review of the followings is requested.
1. The title is grammatically awkward.
2. Overall, English grammar needs correction.
3. It is suggested to add a summary of the overall contents of the review paper in the introduction.
4. Sources for some figures are required (e.g. figure 3)
Author Response
Dear Reviewer 2,
Thank you for your review, which will improve the quality of our article. The manuscript has been revised as recommended and all changes to the text are marked in red. Moreover, the English language was checked by a native speaker.
This review paper describes the overview of polyolefin film extrusion technology. This paper contains more textbook content than an academic paper. However, this paper is thought to be helpful to researchers in the field of polymers because it conveys industrially meaningful content. Therefore, I agree with the publication of this paper. However, a prior review of the followings is requested.
1. The title is grammatically awkward.
Answer: The title has been corrected as recommended
- Overall, English grammar needs correction.
Answer: English language was checked by a native speaker.
- It is suggested to add a summary of the overall contents of the review paper in the introduction.
Answer: A summary of the topics presented in the review has been added to the introduction
- Sources for some figures are required (e.g. figure 3)
Answer: Sources for figures 1 and 3 have been added, other figures have been made by the authors.
Reviewer 3 Report
The results of the work, such as described in the conclusions, are thin. In says that the technologies are efficient, that a range of additives are available for various purposes, low costs for materials and processes and that technologies are widely used. Where are the problems, challenges and the possibilities for improvements? The claim in Abstract "indicating possible technological problems and solutions" is not supported in results and should be improved greatly before publishing.
The text in chapters 1 through 5 if far too long, resembling a summary of several textbooks. What are the important and interesting parts, worth to keep? I would suggest a reduction of the text at least by half.
On line 87 it says that the second polymer in use for production of film is polyethylene. It can be argued with support of reliable statistics, such as Plastics Europe, that PE if by far the largest for packaging applications and the second would be PP. If the statement on line 87 should stand, if needs several more references (not only one).
The tables 1 and 2 should be remade to be similar on properties. Should the stiffness or strength (or both) be included? It would be appropriate to give the type of polymer also in table 2.
The sentence on line 110 needs to be taken away, it is trivial.
The terms used for the different processing types needs to be updated. With support of text books on processing, the term "casting technologies" can be misleading and needs to be revised. Film casting on line 113 is not appropriate, as it can be understood as something different from the described process. The continued text describes an extrusion based process, accordingly it should be named extrusion roll-casting, flat-film extrusion, cast film extrusion, or similar. It can be observed that on line 282 and 285, the term "Extrusion film casting" is used, which is much better.
Author Response
Dear Reviewer 2,
Thank you for your review, which will improve the quality of our article. The manuscript has been revised as recommended and all changes to the text are marked in red. Moreover, the English language was checked by a native speaker.
The results of the work, such as described in the conclusions, are thin. In says that the technologies are efficient, that a range of additives are available for various purposes, low costs for materials and processes and that technologies are widely used. Where are the problems, challenges and the possibilities for improvements? The claim in Abstract "indicating possible technological problems and solutions" is not supported in results and should be improved greatly before publishing.
Answer: In order to improve the quality of work, a paragraph has been added at the end of point 4, which describes modern construction solutions recently used in the cast film extrusion technology.
The text in chapters 1 through 5 if far too long, resembling a summary of several textbooks. What are the important and interesting parts, worth to keep? I would suggest a reduction of the text at least by half.
Answer: Repetitive and obvious sentences were removed from the text. Figures 8, 9 and 10 have been transferred to supplementary materials. Nevertheless, in order to maintain the continuity of the review, important information has been left and has been underlined.
On line 87 it says that the second polymer in use for production of film is polyethylene. It can be argued with support of reliable statistics, such as Plastics Europe, that PE if by far the largest for packaging applications and the second would be PP. If the statement on line 87 should stand, if needs several more references (not only one).
Answer: This sentence has been corrected as recommended.
The tables 1 and 2 should be remade to be similar on properties. Should the stiffness or strength (or both) be included? It would be appropriate to give the type of polymer also in table 2.
Answer: The data in Tables 1 and 2 have been unified.
The sentence on line 110 needs to be taken away, it is trivial.
Answer: The sentence has been removed as recommended
The terms used for the different processing types needs to be updated. With support of text books on processing, the term "casting technologies" can be misleading and needs to be revised. Film casting on line 113 is not appropriate, as it can be understood as something different from the described process. The continued text describes an extrusion based process, accordingly it should be named extrusion roll-casting, flat-film extrusion, cast film extrusion, or similar. It can be observed that on line 282 and 285, the term "Extrusion film casting" is used, which is much better.
Answer: The terms used have been standardized according to the recommendations for "Extrusion film casting" or "cast film extrusion".
Round 2
Reviewer 2 Report
The authors addressed all comments. So I agree to the publication of the manuscript.
Reviewer 3 Report
To my opinion, the manuscript can be printed in the current state.